# Encystation and Stress Responses under the Control of Ubiquitin-like Proteins in Pathogenic Amoebae

**DOI:** 10.3390/microorganisms11112670

**Published:** 2023-10-31

**Authors:** Ascel Samba-Louaka, Elisabeth Labruyère, Mariette Matondo, Marie Locard-Paulet, Jean-Christophe Olivo-Marin, Nancy Guillen

**Affiliations:** 1Université de Poitiers, Centre National de la Recherche Scientifique UMR7267, Laboratoire Ecologie et Biologie des Interactions, TSA51106, 86073 Poitiers, France; 2Institut Pasteur, Biological Image Analysis Unit, Centre National de la Recherche Scientifique UMR3691, Université Paris Cité, 75015 Paris, France; elisabeth.labruyere@pasteur.fr (E.L.); jean-christophe.olivo-marin@pasteur.fr (J.-C.O.-M.); 3Institut Pasteur, Proteomics Core Facility, Mass Spectrometry for Biology Unit, Centre National de la Recherche Scientifique UAR 2024, Université Paris Cité, 75015 Paris, France; mariette.matondo@pasteur.fr; 4Institut de Pharmacologie et de Biologie Structurale, Centre National de la Recherche Scientifique UMR 5089, Université Toulouse III-Paul Sabatier, 31077 Toulouse, France; marie.locard-paulet@ipbs.fr; 5Infrastructure Nationale de Proteomique ProFI—FR2048, 2048 Toulouse, France; 6Institut Pasteur, Centre National de la Recherche Scientifique ERL9195, 75015 Paris, France

**Keywords:** acanthamoeba castellanii, entamoeba histolytica, oxidative stress, endoplasmic reticulum, post-transcriptional modifications

## Abstract

Amoebae found in aquatic and terrestrial environments encompass various pathogenic species, including the parasite *Entamoeba histolytica* and the free-living *Acanthamoeba castellanii*. Both microorganisms pose significant threats to public health, capable of inducing life-threatening effects on humans. These amoebae exist in two cellular forms: trophozoites and cysts. The trophozoite stage is the form used for growth and reproduction while the cyst stage is the resistant and disseminating form. Cysts occur after cellular metabolism slowdown due to nutritional deprivation or the appearance of environmental conditions unfavourable to the amoebae’s growth and division. The initiation of encystation is accompanied by the activation of stress responses, and scarce data indicate that encystation shares factors and mechanisms identified in stress responses occurring in trophozoites exposed to toxic compounds derived from human immune defence. Although some “omics” analyses have explored how amoebae respond to diverse stresses, these studies remain limited and rarely report post-translational modifications that would provide knowledge on the molecular mechanisms underlying amoebae-specific stress responses. In this review, we discuss ubiquitin-like proteins associated with encystation and cell survival during oxidative damage. We aim to shed light on the signalling pathways involved in amoebic defence mechanisms, with a focus on their potential clinical implications against pathogenic amoebae, addressing the pressing need for effective therapies.

## 1. Introduction

Protists of the Amoebozoa clade include several human pathogens, such as *Entamoeba histolytica* and *Acanthamoeba castellanii*, which significantly impact human health. They grow and reproduce in an “active” form called a trophozoite. When subjected to stressful conditions, such as nutrient deprivation, they can efficiently protect themselves by forming cysts in a mechanism called encystation [1,2]. Cysts are characterized by cellular quiescence. Their formation ensures the survival of the species because they resist environmental changes and are easily transmitted to hosts. This is the main contaminating form of these amoebae. Due to their high resilience, cysts also protect from the hosts’ immune system and from biocide treatment.

The free-living *Acanthamoeba castellanii* is found in the environment where it feeds on other microorganisms, such as bacteria, which in turn can use *A. castellanii* as a host or even a vector. The microorganisms found within *A. castellanii* are de facto protected from environmental stresses such as biocides. This way, the amoeba plays a significant role in hosting and disseminating other human pathogens [3]. Thus, in the context of global warming, *A. castellanii* represents a concern for the emergence of unknown pathogens (e.g., giant viruses revived from prehistoric permafrost [4]). *A. castellanii* is also a human pathogen *per se* since it provokes amoebic keratitis, blinding, and granulomatous amoebic encephalitis [3,5]. There are still no specific drugs to treat these diseases [6]. For anti-*A. castellanii* treatments, clinicians use biguanides such as chlorhexidine and polyhexamethylene biguanide, which impair the integrity of cell walls [7,8]; diamidines such as propamidine isethionate which inhibit DNA synthesis [9]; an alkylphosphocholine like miltefosine that seems to have pleiotropic effects [10]; or the cytotoxic agent hydrogen peroxide [11]. These molecules are not always effective, due to the persistence of the cystic stage, and recurrences may occur.

*Entamoeba histolytica*, responsible for amoebiasis, is a eukaryote devoid of mitochondria and a strict parasite of humans. Its cysts contaminate nearly 20% of the population in suburban and rural areas of countries where sanitation practices are compromised. The trophozoite derived from excystation resides in the intestinal lumen where it feeds on bacteria from the microbiota. Within the human gut, this amoeba converts from avirulent to virulent and destroys the muco-epithelial barrier, leading to overproduction of mucus, the killing of human cells, inflammation, and dysentery [12]. The highly mobile trophozoite faces various stresses derived from the host immune response, including oxidative and nitrosative stresses that constitute a challenge for amoebic survival. No vaccine against amoebiasis currently exists. The drug of choice for therapy is metronidazole, which may cause severe side effects in humans and on anaerobic members of the microbiota [13]. The world is widely unprepared for an outbreak of *E. histolytica* due to the lack of a vaccine.

Amoebae submitted to nutrient starvation trigger stress responses [2,13] equivalent to protective responses activated when they are exposed to toxic compounds derived from human immune defences. Omics data show that the regulation of protein abundance (i.e., Hsp20 in *A. castellanii* or Hsp70, chitinase, and cyst wall proteins in *E. histolytica*) [14,15] and upregulation of gene expression by RNAs (i.e., tRNA, anti-sense RNA, and mRNA) are common to encystation and oxidative stress [1,16,17,18,19]. Despite the importance of pathogenic amoebae in ecology and medicine, the stress responses of *Acanthamoeba* spp. and *Entamoeba* spp. remain under-investigated.

In this review, we discuss the current state of our knowledge on (i) the stress responses common in eukaryotes and (ii) the specific mechanisms of stress in pathogenic amoebae during encystation and under attack of reactive species of oxygen. We then give significant emphasis to protein post-translational modifications (PTMs), in particular those involving ubiquitin-like modifiers (UBLs), of which the importance in stress has been recently discovered [20]. Identifying these molecular mechanisms will enhance our understanding of the signalling pathways involved in amoebic defence and should provide avenues for clinical intervention against pathogenic amoebae.

## 2. Stress Responses Common in Eukaryotes

The mechanisms of stress responses have been largely studied in eukaryotic cells, although bacteria possess protein components of these pathways. Diverse protective responses to stress occur depending on the severity of the stress stimulus. These protective responses to stress include specific signalling pathways that can occur simultaneously in cells undergoing damage. Failure to activate these responses fuels cell death pathways. Here, we briefly summarize some features of these responses according to the extensive research on eukaryotic cells.

The DNA damage response is caused by the exposure of cells to chemotherapeutic agents, irradiation, or environmental genotoxic agents, such as polycyclic hydrocarbons or ultraviolet (UV) light, that lead to DNA breaks [21]. Double-strand breaks and single-strand breaks are lesions that trigger the activation of the DNA damage response, thereby activating DNA repair pathways [21].

The oxidative stress response is triggered by reactive oxygen species (ROS), which are the most important pro-oxidants produced during metabolic reactions. When cells experience changes in the balance between pro-oxidants and antioxidants, there is an increase in oxidative stress (OS). ROS oxidize macromolecules (i.e., nucleic acids, proteins, lipids, and sugars) and cause the loss of their function. Glucose deprivation increases ROS levels [22] through the activity of AMP-activated protein kinase (AMPK), which reprograms metabolic pathways in order to maintain energy balance in cells [23]. If prolonged, starvation causes autophagy and, subsequently, encystation. ROS are generated in the cytosol and in several subcellular endomembrane systems, including the plasma membrane, peroxisomes, the membranes of mitochondria, and the endoplasmic reticulum (ER) [24].

The autophagic response occurs in cells exposed to metabolic or therapeutic stresses, such as growth factor deprivation, inhibition of the serine/threonine kinase mTOR (mechanistic Target Of Rapamycin, which coordinates proliferation and metabolism with the environmental conditions) signalling, limitation of nutrients, inhibition of proteasomal degradation, accumulation of intracellular calcium, and ER stress. Autophagy is regulated by a family of autophagy-related genes (ATGs) and starts with the sequestration of damaged organelles or cytosolic components into double-membrane autophagosomes, followed by fusion with lysosomes where the damaged products are degraded to produce energy during stress [25].

The heat shock response is initiated by increased levels of heat shock proteins (HSPs), which act as molecular chaperones to prevent protein misfolding and aggregation. HSPs are also upregulated in response to various stressors, including ER stress, oxidative stress, and nutrient deprivation [26,27]. The unfolded protein response (UPR) is due to the accumulation of unfolded proteins in the ER [28]. UPR promotes cell survival by improving the balance between protein loading and ER folding capacity. Misfolded peptides are exported from the ER and degraded either by the proteasome (ER-associated degradation, ERAD) or by the lysosome (ER-to-lysosome-associated degradation, ERLAD). In ERAD, the ubiquitin–proteasome system is a component of the intracellular protein degradation mechanism.

In the context of this review, the question arises of what is known about stress responses in pathogenic amoebae.

Although research on this topic has remained rather scarce, here we will consider referenced data from pathogenic amoebae affecting humans, such as the HM-1:IMSS strain of *E. histolytica* and the Neff strain of *A. castellanii*. Concerning encystation, we will add data from the strain IP1 of *Entamoeba invadens*, the reptilian intestinal parasite that encysts in vitro.

## 3. Mechanisms of Stress in Pathogenic Amoebae during Encystation

Encystation is a differentiation process triggered in response to prolonged nutritional deprivation stress. From the trophozoite state, the pathogenic amoebae differentiate into dormant resistant cysts. Our unpublished data indicate that early morphology changes leading to encystation can be followed in living trophozoites using video microscopy, as shown in Figure 1 with *A. castellanii*. This process requires the total reshaping of trophozoites that become rounded. It is known that cyst-wall components are then synthesized [29] and transported toward the plasma membrane in specific encystation vesicles containing fibrillar material [30,31,32]. The membrane trafficking system is also modulated since secretory pathway- and trans-Golgi network-endosome-related genes are upregulated during cyst formation [33]. Gradual dehydration reduces trophozoite volume by approximately 80% following retraction of the cytoplasm from the cyst wall [34]. Cell signalling under encystation stress increases cAMP levels and activates protein kinase A, known to widely control amoebozoan encystation. At the biochemical level, glycogen is rapidly degraded for cyst-wall synthesis. Sugar polymers that make fibrils include beta-1,4-linked GlcNAc (chitin) and beta-1,4-linked glucose (cellulose). *Entamoeba* makes chitin, while the *Acanthamoeba* cyst wall is mainly composed of cellulose. AMPK, the enzyme that maintains ATP homeostasis [23], has been shown to be involved in encystation, as well as in heat stress and OS of *Entamoeba* [35], and has homologs in *Acanthamoeba* that could also be involved in stress responses.

## 4. Mechanisms of Stress in Pathogenic Amoebae during Accumulation of Reactive Oxygen Species 

Increase of ROS levels (i.e., peroxides, superoxides, hydroxyl radicals, and singlet oxygen), play important roles in cell signalling and homeostasis, and cause damage to the cells through OS. Mitochondria-containing amoebae (e.g., *Acanthamoeba*) react to ROS (as other eukaryotes) via oxidative phosphorylation through the activity of antioxidant enzymes such as superoxide dismutase, catalase, glutathione peroxidase, and glutathione reductase [36]. However, high levels of ROS damage the membrane structure by targeting lipids and proteins. The literature reports several molecules able to regulate the *Acanthamoeba* OS response. Exposition to pure oxygen has been shown to increase production in ROS [37], whereas chlorhexidine increases the ROS levels in *Acanthamoeba* through a reduction in the activity of antioxidant enzymes such as the superoxide dismutase (SOD) and the flavin adenine dinucleotide–fumarate reductase (NADH-FRD) [38]. Conversely, statins were shown to provoke a collapse in the mitochondrial potential and ATP levels, cytoskeleton disassembly, and ROS generation [39]. The ability of *A. castellanii* to maintain a redox balance probably results from a large panel of paths. It can also efficiently use an uncoupling of respiratory substrate oxidation from ATP synthesis [40], a regulation of the expression of the superoxide dismutase, of the catalase, and of both the thioredoxin and the glutathione systems [36,41,42,43].

Amitochondriate amoebae (e.g., *Entamoeba*) respond to OS using a large arsenal of redox pathways [44], since the basic antioxidant defence mechanisms of eukaryote cells (such as reduced glutathione, catalase, and glutathione-recycling enzymes) lack in *Entamoeba* and their major antioxidant is L-cysteine, a sulphur-containing amino acid [45]. In the absence of mitochondria, *Entamoeba* species retain an organelle derived from mitochondria called mitosome devoid of any organellar genome. It is the site of sulfolipid synthesis, and certain enzymes of this pathway are involved in encystation [46,47]. The mitochondrial eukaryote iron–sulphur assembly system is replaced in *E. histolytica* by a nitrogen fixation system whose coding genes were acquired through lateral gene transfer from ε-proteobacteria [48]. Iron–sulphur (Fe–S) clusters are ubiquitous cofactors that mediate a multitude of functions including electron transfer and redox reactions. In mitosomes, the *Entamoeba* hydroxyperoxyde detoxification enzyme rubrerythrin contains Fe-S clusters [49]. Whereas the eukaryotic assembly of Fe-S cofactors is mediated by proteins inherited from the original mitochondrial endosymbiont, in *E. histolytica*, the analysis of two genes encoding critical Fe-S cluster (Isc) biosynthetic proteins suggests a lateral acquisition from epsilon proteobacteria [48,49,50]. In *Entamoeba*, during OS, there is an activation of oxidation of Fe-S clusters inhibiting glycolysis-related enzymes: dismutation by superoxide dismutase to remove superoxide radical anion O_2_^−^ and peroxiredoxins to remove H_2_O_2_ [51]. The inhibition of glycolysis triggers encystation, leading to the redirection of metabolic flux towards glycerol production [52] and chitin wall biosynthesis [53]. The interplay between components of mitosomes, ER, and the membrane of intracellular vesicles sites is under investigation; recently, the transmembrane mitosomal protein1 was identified in mitosome–endosome contact sites [54].

## 5. UPR and Endoplasmic Reticulum Stress Responses Identified in Pathogenic Amoebae

*Entamoeba histolytica* maintain intracellular hypoxia in human oxygenated tissues and cellular homeostasis during the host immune defence attack. The trophozoites stimulate cysteine synthase activity, inhibit glycolysis, and induce an UPR [55]. Considering that no classic ER structure has been described in *E. histolytica*, studies regarding UPR have remained scarce. In mammalian cells, UPR is activated in parallel by three ER transmembrane sensors: the inositol-requiring enzyme 1 (IRE1), the protein kinase RNA-like ER kinase (PERK), and the activating transcription factor 6 (ATF6). This leads to an increase in the transcription of chaperone-encoding genes, e.g., Hsp70, whose over-expression attenuates the UPR. In *E. histolytica*, UPR- and ER-associated degradation are interconnected, as shown by the upregulation of Ubc7 and IRE1 upon the accumulation of misfolded proteins in the ER [56]. PERK is activated by misfolded proteins and phosphorylates the alpha subunit of eukaryotic initiation factor-2 (eIF2α), causing a reduction in protein synthesis and the preferential translation of mRNA coding for from stress-related genes. *E. histolytica’s* genome does not encode obvious orthologues of PERK or ATF6; however, the phosphorylation of eIF2α occurs after serum starvation, heat shock, OS [57], and ER stress [58]. In *E. invadens*, silencing the gene encoding EieIF2α (homologous with eIF2α) leads to increased sensitivity to OS and a reduction in encystation rate [59]. These data suggest that there could be an incomplete UPR pathway in *Entamoebae* [57]. Virulent *E. histolytica* cultured under OS display massive upregulation of genes encoding HSPs [13] and diverse transcriptome analysis of *E. histolytica*, highlighting that the Hsp70-A-encoding gene is upregulated following OS and nitrosative stress (NO), and during the formation of hepatic abscesses; therefore, the inhibition of Hsp70-A blocks virulence [60]. The treatment of *E. histolytica* with NO inhibits glycolysis, enhances cysteine synthase activity, and dramatically provokes extensive ER fragmentation [61] (Figure 2). As in other systems [62], ER stress and OS in *Entamoebae* seem to be strongly correlated, although the above-described specificities give perspectives for new drug discovery.

While the antioxidant activity of *A. castellanii* seems of interest to researchers, less is known about its interplay with the other stress response pathways. For example, OS and ER stress are two important phenomena that accentuate each other in a positive feed-forward loop [63]. *A. castellanii* copes with exposure to ROS inducers by modulating the amount of HSP70/HSP60 [64]. To our knowledge, the UPR that results from ER stress has never been investigated in *A. castellanii*. Discovery of the ER-mitochondria encounter structure in *A. castellanii* is a window that could help to address ER stress [65]. Another aspect that is worth investigation is the link between OS and the proteasome function. Indeed, the latter is important for redox balance [66]. The importance of the proteasome during a stress response such as encystation was already reported [18,67], but no study has explored the expression of proteasome-coding genes or the proteasome activity upon OS.

## 6. Ubiquitin, Ubiquitin-like Protein Modifications, and Proteasome Activities in Pathogenic Amoebae

Stress leads to ubiquitin and ubiquitin-like protein modifications that target protein degradation by the proteasome. Finely regulated physiological processes rely on post-translational protein modification with ubiquitin (Ub) and ubiquitination constitutes a major source of proteome regulation during stresses. The covalent attachment of Ub (a 76-residue peptide) is called monoubiquitination and can be followed by the addition of more Ub: multiubiquitination or polyubiquitination of proteins. The degree of ubiquitination accounts for diverse regulations of the substrate. For example, polyubiquitinated proteins are typically degraded by the 26S proteasomal complex, whereas mono/multiubiquitinated proteins are not degraded and can be involved in endocytosis, membrane trafficking, regulation of kinase signalling, DNA repair, and chromatin regulation. Ubiquitination involves a sequential enzymatic cascade constituted by an activating enzyme (E1), a conjugating enzyme (E2), and a ligating enzyme (E3) that covalently bind Ub to the target protein. Moreover, there are specific peptidases (deubiquitinases) that reverse the action of E3 ligases by removing the Ub from the proteins. Comparative genomics analysis suggests that the Ub system had a pre-eukaryotic origin [68]. Ub, ER-associated Ub-conjugating enzymes, and genes potentially involved in ER-associated degradation have been identified in *E. histolytica* [69] and in *Acanthamoeba* [70]. Figure 3 shows that many genes are currently annotated with functions related to protein ubiquitination in the reference genomes of *Entamoeba histolytica* (*Entamoeba histolytica* HM-1:IMSS) and *Acanthamoeba castellani* (*Acanthamoeba castellani* str. Neff), and in the genome of *Entamoeba invadens* (*Entamoeba invadens* IP1).

Other PTM signalling pathways known as UBL systems have been characterized by their similarity with the tertiary structure of Ub (a β-grasp fold). The UBL family includes neural precursor cell-expressed developmentally downregulated 8 (NEDD8), small ubiquitin-related modifier (SUMO), ubiquitin-fold modifier 1 (UFM1), ubiquitin-related modifier 1 (URM1), autophagy-related proteins 8 and 12 (ATG8 and ATG12), interferon-stimulated gene 15 (ISG15), and human leukocyte antigen-F adjacent transcript 10 (FAT10). Despite their structural similarities with the Ub system, UBL systems exhibit distinct enzymatic cascade structures with unique sets of enzymes. The understanding of the mechanistic aspects and biological roles of UBL pathways has significantly advanced in yeast, plant, and mammalian cells. A rich variety of UBLs have been found in parasitic protozoa [71], with the exception of ISG15 and FAT10, which are involved in the immune and stress responses of multicellular organisms. All UBLs, with the exception of metazoan-specific UBLs and UFM1, are encoded by *Entamoeba* species [69]. In the context of this review, we focus on important components of stress responses involved in autophagy, which is important for encystation and ER stress/UPR involved in OS.

### 6.1. Autophagy-Related Proteins in Pathogenic Amoebae

Autophagy is an intracellular degradation and recycling system that is ubiquitous in eukaryotic cells. This evolutionarily conserved process is a cellular response to starvation and stress. Excess or dysfunctional proteins, organelles, or pathogens are destroyed after sequestration into double-membrane vesicles called autophagosomes that are fused with lysosomes for turnover. ATG proteins control the biogenesis of autophagosomes. Among the several proteins involved in the completion of autophagy, ATG8, which binds to autophagic membranes through conjugation to phosphatidylethanolamine and phosphatidylserine, represents a hallmark of autophagy [72].

In *A. castellanii*, upon starvation, the protein ATG8 was shown to contribute to autophagy through the binding of the autophagosome membrane, where it mediates the recruitment of different cargo molecules into autophagosomes. The activity of the *A. castellanii* ATG8 was confirmed via genetic complementation in *Saccharomyces cerevisiae* [73]. The protein ATG3, which is involved in the ATG8 conjugation system, was also reported to be involved in the encystation of *A. castellanii* [74].

In *E. histolytica*, ATG8 is involved in the biogenesis of endosomes, phagocytosis, and trogocytosis (at least maturation of phago/trogosomes) [75,76]. The second conjugation system ATG5/12 may not be functional or highly diverged as ATG12 is missing in the *Entamoeba* genome [77]; nevertheless, the ATG12 conjugation system seems present in *A. castellanii*, where it is essential for encystation [78]. ATG16, which is important for the correct localization of the complex ATG5/12 conjugate to the pre-autophagosomal structure [79], was also shown to play a role in autophagosome formation and in *Acanthamoeba* encystation [80].

### 6.2. UFM1, URM1 in Amoebae, a Terra Incognita

Ufmylation is a cascade reaction involving an E1-E2-E3 system, based on the Ubiquitin-Fold Modifier 1 (UFM1) conjugation system. UFM1 is one of the newly identified UBL molecules conserved among nearly all eukaryotes, except yeasts and other fungi [81]. UFM1 is known to play a role in various cellular processes, including protein quality control, cellular stress responses, regulation of protein stability, and regulation of ER activity. The genetic disruption of Ufmylation genes strongly induces ER and OS, and UFM1 genes are transcriptional targets of UPR, suggesting a functional relationship between Ufmylation and ER proteostasis [82,83]. However, the UFM1 substrates also include proteins involved in DNA damage repair, hematopoiesis, and autophagy [84]. Thus, Ufmylation components appear to be essential in several stress responses [84]. For instance, UFBP1 (UFM1-binding and PCI domain-containing protein 1) has been shown to induce proteasomal degradation of the master OS-response transcription factor Nrf2 (nuclear factor erythroid-2-related factor 2) [85]. While more insights are continually being published regarding ufmylation in human cells, nothing is known yet of its presence and/or function in amoebae. Although evidence suggests the absence of UFM1 in *Entamoeba*, some observations suggest that ufmylation exists in *A. castellanii* (Figure 3) [18,86]. The localization of ufmylated proteins within amoebae is also a subject worthy of investigation. If ufmylated proteins are mainly described in ER and in the nucleus, in certain microorganisms, such as *Leishmania donovani*, UFM1 and its conjugation machinery are associated with the mitochondria [87]. Determining the patterns of ufmylated proteins within amoebae is an area that warrants further exploration.

Urmylation results from the conjugation of the Ub-related modifier 1 (URM1) to target proteins. URM1 is the oldest UBL system [88], conserved from bacteria to humans [89]. URM1 undergoes activation via C-terminal thiocarboxylation. Thiolated-URM1 has a dual function, firstly as a sulphur carrier protein for thiolation of tRNA anticodons, and secondly as a UBL modifier in many cells [88,90]. Thus, URM1 interacts with proteins associated with OS, ubiquitination, nuclear export, tRNA modification, RNA binding, and organism longevity [90]. Peroxiredoxins are highly conserved thiol-dependent proteins which detoxify cells from ROS; from these, 2-Cys peroxiredoxins (Ahp1, Prx5) undergo urmylation during OS [88,89,91]. In addition to URM1, only UBA4, which shows similarity to E1-activating enzymes, is known in urmylation. URM1 is present in *E. histolytica* according to phylogenetic analyses (Figure 3) and can be expected to be discovered in *Acanthamoeba*. However, until now, there have been no experimental data connecting URM1 and encystation or OS responses of pathogenic amoebae. Nevertheless, we can anticipate that, according to other systems, URM1 may be present in *Entamoeba* peroxisomes as seven peroxiredoxins are associated to this organelle [92], a hypothesis that deserves testing.

## 7. Conclusions and Perspectives

*Entamoeba* and *Acanthamoeba* seem well equipped to respond to cellular stresses. However, the mechanisms involved in their different stress responses have not been equally investigated. *Acanthamoeba* ER stress has been overlooked by researchers, whereas the study of *E. histolytica* encystation in vitro is still facing experimental limitations. Therefore, large knowledge gaps remain in our understanding of proteome changes during amoebae adaptation to stressful conditions. There is no doubt that the application of high-sensitivity mass-spectrometry approaches, such as proteomics, phosphoproteomics, and interactomics, combined with newly developed informatics strategies will help to fill these gaps. Furthermore, even if *E. histolytica* has twice fewer protein coding genes than *A. castellanni*, it is possible to compare their proteomes since they share 56% of their ortholog groups [93]. From a cell biology point of view, it is important to precisely determine when ER proteostasis or commitment of encystation occurs, as well as to map the intracellular traffic of UBL-modified proteins. Live cell imaging and computerized image analysis approaches should correlate the localization of these proteins in intracellular compartments with cell morphological changes during stress responses and the encystation process. The growing interest in UBLs is opening the way to new therapeutic targets against these two organisms for which the set of available drugs remains limited.

## Figures and Tables

**Figure 1 microorganisms-11-02670-f001:**
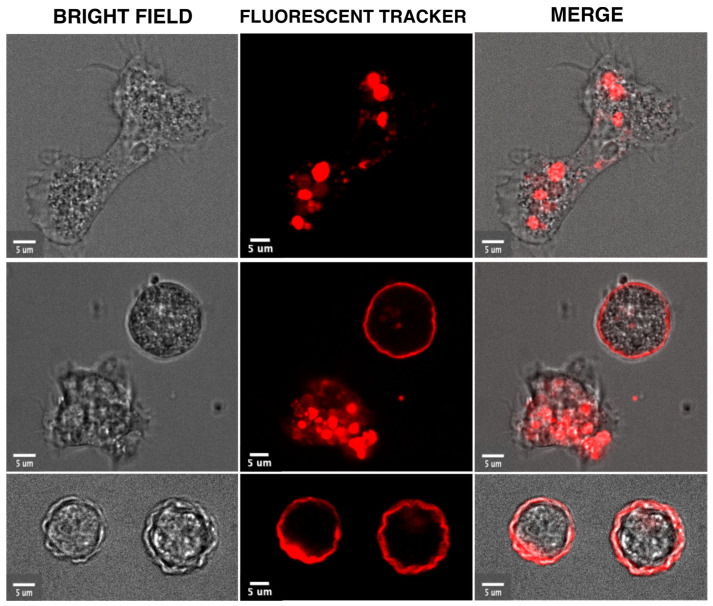
Morphological and intracellular changes visualized in video microscopies acquired during the encystation of *A. castellanii.* The trophozoites were incubated in media containing a red fluorescent tracer (Cell Tracker™ Red) which accumulates in cytoplasm and intracellular compartments. The unpublished micrographs show the elongated trophozoite in a rich medium (upper panel) and a rounded trophozoite stressed by food deprivation next to a cyst (middle panel) and cysts (third panel). These cellular forms were visualized by confocal microscopy with bright field (left panel) or fluorescence light (middle panel). The two micrographs are overlaid in the right panel. Bar corresponds to 5 μm.

**Figure 2 microorganisms-11-02670-f002:**
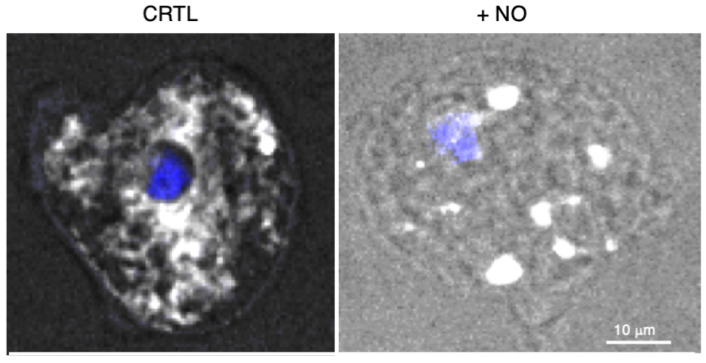
*E. histolytica* submitted to stress by the addition of nitric oxide (NO). The endoplasmic reticulum was followed by confocal microscopy, recovering the fluorescence of the green fluorescent protein-FLAG construct linked to the ER retention motif KDEL (white signal obtained with an anti-FLAG antibody) and the nucleus (4′,6-diamidino-2-phenylindole, DAPI in blue). Protocol details are in reference [58]. The fluorescence is homogenous in resting condition (left panel), whereas ER fission is observed as vesicular patterns in the presence of NO (right panel). Bar scale corresponds to 10 μm.

**Figure 3 microorganisms-11-02670-f003:**
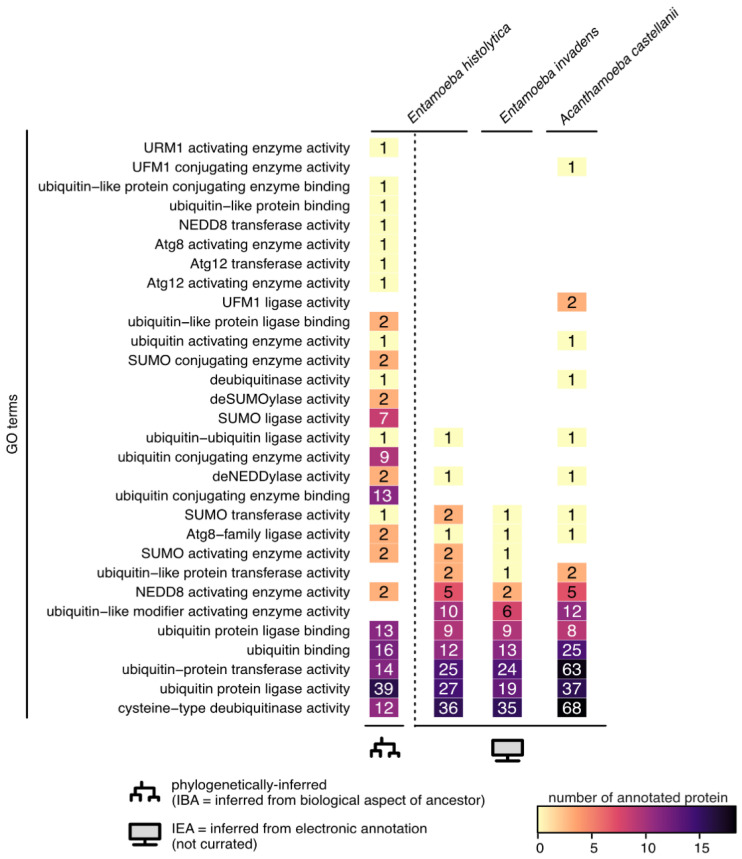
Ubiquitin-like associated function in pathogenic amoebae. Number of protein accessions annotated with ubiquitin-like function in the reference genomes of *Entamoeba histolytica* (*Entamoeba histolytica* HM-1:IMSS-taxon 294381), *Entamoeba invadens* (*Entamoeba invadens* IP1-taxon 370355), and *Acanthamoeba castellani* (*Acanthamoeba castellani* str. Neff-taxon 1257118). GO terms were retrieved from the QuickGo website (ebi.ac.uk/QuickGO) on the 9th of September 2023. *E. histolytica* HM-1:IMSS is the only taxon with phylogenetically inferred biological functions (IBA); all the others only have automatic computationally inferred annotations (IEA). Gene identities are in Appendix A.

## Data Availability

Data sharing is not applicable.

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
