# Peer review of "Encystation and Stress Responses under the Control of Ubiquitin-like Proteins in Pathogenic Amoebae"

_microorganisms, 2023, doi:10.3390/microorganisms11112670_

Round 1
Reviewer 1 Report
The review is very interesting: an actual topic has been chosen and a good presentation of the available data is offered to the readers; it is easy to read. The topic is very actual especially in the époques of rising of opportunistic pathogens and the urgent need to develop adequate systems for control and inhibition of parasite growth and development.
I have few corrections and suggestions:
Line 2
Consider reformatting: I would not recommend using hyphens in the title
Line 24
The trophozoite stage is one of growth and reproduction – I suggest: The trophozoite stage is one for growth and reproduction
the cyst stage is the resistant and contaminating form – I suggest: disseminating form
Line 64
Entamoeba should not be called an ancient organism; it belongs to an advanced (but secondary reduced and simplified) clade within Amoebozoa.
Line 114
Decipher mTOR: the mechanistic Target Of Rapamycin, a serine/threonine protein kinase, which coordinates parasite proliferation and metabolism with the environmental conditions (e.g. nutrients from the host)
Line 130-131
The diverse mechanisms of stress responses have been largely studied in eukaryotic cells although bacteria, protozoans, fungi, and yeast possess protein components of these pathways. - the phrase is formulated incorrectly: protozoa, fungi and yeast are also eukaryotic organisms.
Line 181-184
The mitochondrial eukaryote iron–sulphur assembly system is replaced in E. histolytica by nitrogen fixation system whose coding genes were acquired by lateral gene transfer from ε-proteobacteria [45]. In Entamoeba, there is an activation of oxidation of iron-sulphur clusters inhibiting glycolysis-related enzymes.
I would recommend to provide more details on the story about horizontal transfer of iron–sulphur clusters from ε-proteobacteria. Currently the description of this story in the ms is too short and not fully understandable, and in some places even confusing. Consider revising.
Line 273
Wouldn't it be more correct to write: ubiquitin-associated functions?
I noted a few points that apparently required polishing. They are indicated below in the comments.
Author Response
Reviewer 1
The review is very interesting: an actual topic has been chosen and a good presentation of the available data is offered to the readers; it is easy to read. The topic is very actual especially in the époques of rising of opportunistic pathogens and the urgent need to develop adequate systems for control and inhibition of parasite growth and development.
Authors: We thank the reviewer for the positive comment.
I have few corrections and suggestions:
Line 2
Consider reformatting: I would not recommend using hyphens in the title
Authors: this has been corrected.
Line 24
The trophozoite stage is one of growth and reproduction – I suggest: The trophozoite stage is one for growth and reproduction
Authors: We have modified accordingly.
the cyst stage is the resistant and contaminating form – I suggest: disseminating form
Authors: this has been changed accordingly.
Line 64
Entamoeba should not be called an ancient organism; it belongs to an advanced (but secondary reduced and simplified) clade within Amoebozoa.
Authors: the word “ancient” has been removed.
Line 114
Decipher mTOR: the mechanistic Target Of Rapamycin, a serine/threonine protein kinase, which coordinates parasite proliferation and metabolism with the environmental conditions (e.g. nutrients from the host)
Authors: we thank the reviewer for the suggestion that has been included in the manuscript.
Line 130-131
The diverse mechanisms of stress responses have been largely studied in eukaryotic cells although bacteria, protozoans, fungi, and yeast possess protein components of these pathways. - the phrase is formulated incorrectly: protozoa, fungi and yeast are also eukaryotic organisms.
Authors: we apologize for the mistake, we removed “protozoans, fungi and yeast” from the sentence.
Line 181-184
The mitochondrial eukaryote iron–sulphur assembly system is replaced in E. histolytica by nitrogen fixation system whose coding genes were acquired by lateral gene transfer from ε-proteobacteria [45]. In Entamoeba, there is an activation of oxidation of iron-sulphur clusters inhibiting glycolysis-related enzymes.
I would recommend to provide more details on the story about horizontal transfer of iron–sulphur clusters from ε-proteobacteria. Currently the description of this story in the ms is too short and not fully understandable, and in some places even confusing. Consider revising.
Authors: we thank the reviewer for the comment. We replaced the former sentence “The mitochondrial eukaryote iron–sulphur assembly system is replaced in E. histolytica by nitrogen fixation system whose coding genes were acquired by lateral gene transfer from ε-proteobacteria.” By “Iron–sulfur (Fe–S) clusters are ubiquitous cofactors that mediate a multitude of functions including electron transfer and redox reactions. In mitosomes, the Entamoeba hydroxyperoxyde detoxification enzyme rubrerythrin contains Fe-S centers (Barbora Maralikova et al., 2010 Cellular Microbiology). Whereas the eukaryotic assembly of Fe-S cofactors is mediated by proteins inherited from the original mitochondrial endosymbiont, in E. histolytica, the analysis of two genes encoding critical Fe-S cluster (Isc) biosynthetic proteins suggest a lateral acquisition from epsilon proteobacteria (Barbora Maralikova et al., 2010 Cellular Microbiology; Mark van der Giezen et al., 2004 BMC Evolutionary Biology; [45])". We also order the paragraph in a more didactic form.
Line 273
Wouldn't it be more correct to write: ubiquitin-associated functions?
Authors: thank you, the title of the figure 3 has been changed into “Ubiquitin-like associated function in pathogenic amoebae”.
Reviewer 2 Report
Encystation and stress responses under the control of ubiquitin-like proteins in pathogenic amoebae
Comments
In this review, the authors explore the behavior of pathogenic amoeba, but mainly two amoebae, Acanthamoeba castellanii, and Entamoeba histolytica, showing the reaction of the organisms when infected by these parasites.
First, a review of this topic is important. However, this work is a review and, as such, should be better organized. The authors address a topic, then write a subtitle, and we don't understand which amoeba it is about. There is a lack of comparisons between the findings in the two amoebae in the abstract. Many sentences are loose, and we don't know which cell they refer to. There is a lack of more images on the topic and more care in presenting the few images. We don't even know whether they are previously published images or whether they are from these authors or others. The few images presented have poorly made magnification bars and no standardization. Sometimes, there are figures without indications of value, while the value is invisible in others.
Minor comments:
Lines 86-87. We give significant emphasis to protein post-translational modifications (PTMs), in particular those involving ubiquitin-like modifiers (UBLs), which importance in stress has been recently discovered."
Please add a reference here.
Line 92: "... the cell can cope by triggering an"- Please explain which cell you refer to.
Line 111: "Endoplasmic Reticulum", please change by endoplasmic reticulum.
Line 139: "into spheres: cytoplasmic projections such as pseudopodia,." This sentence is confusing. Please re-write.
Line 192: ER and the cytosolic membrane system"- What do you mean? What is a cytosolic membrane system?
The authors do not mention from who/where are the figures presented in the review. Were they previously published or unpublished or not? Please, add this information.
In all figures, the bars are not well presented. Either are invisible or without the numeric value. Please, standardize.

Author Response
Answers to the queries from reviewers of the article: microorganisms-2655752
"Encystation and stress responses under the control of ubiquitin -like proteins in pathogenic amoebae" by Samba-Louaka et al., submitted to Microorganisms.
Reviewer 2
In this review, the authors explore the behavior of pathogenic amoeba, but mainly two amoebae, Acanthamoeba castellanii, and Entamoeba histolytica, showing the reaction of the organisms when infected by these parasites. First, a review of this topic is important. However, this work is a review and, as such, should be better organized.
Authors: we thank the reviewer for the comments, and we have modified the document according to his/her criticisms.
The authors address a topic, then write a subtitle, and we don't understand which amoeba it is about.
Authors: we precise the genus and species of amoebae discussed in this review. In the text of each chapter, we have carefully indicated the identity of the amoebae concerned by the topic/paragraph.
There is a lack of comparisons between the findings in the two amoebae in the abstract. Many sentences are loose, and we don't know which cell they refer to.
Authors: the subject of this review is broad, and the main text consists of the comparison between the two amoebae; our comments has been organized based on the extensive data from other eukaryotes. We consider that the main objective of this work is clearly stated in the abstract, namely, to understand how amoebae respond to various stresses, including those due to encystation or oxidative stress. The reviewer did not specify which sentences he/her believed were truncated, thus we couldn't correct. As already stated, we have carefully indicated the identity of the amoebae concerned by the topic/paragraph.
There is a lack of more images on the topic and more care in presenting the few images.
Authors: Because the topic of UBLs functions in pathogenic amoebae is poorly explored we cannot provide more images.
We don't even know whether they are previously published images or whether they are from these authors or others. The few images presented have poorly made magnification bars and no standardization. Sometimes, there are figures without indications of value, while the value is invisible in others.
Authors: these images are from our own laboratories, there is three figures and one table which is standard for a review. We have provided precisions in their legends.
Minor comments:
Lines 86-87. We give significant emphasis to protein post-translational modifications (PTMs), in particular those involving ubiquitin-like modifiers (UBLs), which importance in stress has been recently discovered."
Please add a reference here.
Authors: the following reference has been inserted within the text (Dragana Ilic et al., 2022 Seminars in Cell & Developmental Biology).
Line 92: "... the cell can cope by triggering an"- Please explain which cell you refer to.
Authors: this sentence has been removed and the paragraph has been rewritten as “The diverse mechanisms of stress responses have been largely studied in eukaryotic cells although bacteria possess protein components of these pathways. Diverse protective responses to stress occur depending on the severity of the stress stimulus.”.
Line 111: "Endoplasmic Reticulum", please change by endoplasmic reticulum.
Authors: done
Line 139: "into spheres: cytoplasmic projections such as pseudopodia,." This sentence is confusing. Please re-write.
Authors: We sincerely apologize, the sentence has been rephrased as “This process requires the total reshaping of trophozoites that become rounded, with cyst-wall components that are synthesized [28] and transported toward the plasma membrane in specific encystation vesicles containing fibrillar material [29–31]”.
Line 192: ER and the cytosolic membrane system"- What do you mean? What is a cytosolic membrane system?
Authors: we apologize for the misunderstanding. We have changed the entire sentence as follow: "The interplay between components of mitosomes, ER and the membrane of intracellular vesicles sites are under investigation."
The authors do not mention from who/where are the figures presented in the review. Were they previously published or unpublished or not? Please, add this information.
Authors: precisions were added in the legends
In all figures, the bars are not well presented. Either are invisible or without the numeric value. Please, standardize.
Authors: The bars and numeric values are visible in the figures and numeric values are in addition added in the legend
Reviewer 3 Report
This paper discuss two important pathogenic amoebae species Entamoeba histolytica and Acanthamoeba castellanii, and the molecular mechanisms of their stress responses and survival. Undoubtedly, the review is valuable and can become a solid background for further studies focusing on functional genomics and clinical investigations against these pathogens, however I think the review could be better structured and clear in some passages.
My main concern is the missing link between the stated objectives in the introduction and the flow of the text. For example: In the Introduction (L83-85) it is stated that “In this review we discuss the current state of our knowledge on: (i) the stress responses common between amoebae and other organisms, (ii) the specific mechanisms of stress in amoebae during encystation and under attack of reactive species of oxygen. However the following text (especially part 2) does not fully reflect the objective “(i) the stress responses common between amoebae and other organisms”. I would start the part 2 with the sentence in L. 130-131 “The diverse mechanisms of stress responses have been largely studied in eukaryotic cells ….. such as DNA damage response, oxidative stress response, autophagic response, heat shock response, etc…… and then move to part (ii) the specific mechanisms of stress in amoebae….with the focus on specific gene/protein functional groups. I also did not feel a comparison of stress response related mechanisms of amoebae and other organisms as it was indicated in the Objective (I). Perhaps creating a small comparative table indicating the main stress response mechanisms, related functional genes common for different group of organisms can be useful.
I am also confused about information included in the figures. From where is this information coming? If it is the result of your own study then the legends should include more details about methodology. If these are not your results then the clear reference should be included in the legends.
More specifically:
Figure 3. Genome accession numbers must be indicated.
L70-80 in the Introduction: I am not sure if I understand this sentence, it is very abstract and not informative => Omics data show the regulation of protein abundance [14,15] and gene regulation (i.e., tRNA, anti-sense RNA, mRNA) are common to encystation and oxidative stress [1,16–19]. => I suggest to be here more specific, which proteins, which genes? Are you talking about gene up or down regulation?
L86-87: We give significant emphasis to protein post-translational modifications (PTMs), in particular those involving ubiquitin-like modifiers (UBLs), which importance in stress has been recently discovered => provide reference
L91. 2. Shared stress responses enabling cell survival => Shared? Why shared? May be better to say combined?
L250 => is this about amoebae or in general?
see my comments above (L70-80, L91)
Author Response
Answers to the queries from reviewers of the article: microorganisms-2655752
"Encystation and stress responses under the control of ubiquitin -like proteins in pathogenic amoebae" by Samba-Louaka et al., submitted to Microorganisms.
Reviewer 3
This paper discuss two important pathogenic amoebae species Entamoeba histolytica and Acanthamoeba castellanii, and the molecular mechanisms of their stress responses and survival. Undoubtedly, the review is valuable and can become a solid background for further studies focusing on functional genomics and clinical investigations against these pathogens, however I think the review could be better structured and clear in some passages.
My main concern is the missing link between the stated objectives in the introduction and the flow of the text. For example: In the Introduction (L83-85) it is stated that “In this review we discuss the current state of our knowledge on: (i) the stress responses common between amoebae and other organisms, (ii) the specific mechanisms of stress in amoebae during encystation and under attack of reactive species of oxygen. However the following text (especially part 2) does not fully reflect the objective “(i) the stress responses common between amoebae and other organisms”. I would start the part 2 with the sentence in L. 130-131 “The diverse mechanisms of stress responses have been largely studied in eukaryotic cells ….. such as DNA damage response, oxidative stress response, autophagic response, heat shock response, etc…… and then move to part (ii) the specific mechanisms of stress in amoebae….with the focus on specific gene/protein functional groups. I also did not feel a comparison of stress response related mechanisms of amoebae and other organisms as it was indicated in the Objective (I). Perhaps creating a small comparative table indicating the main stress response mechanisms, related functional genes common for different group of organisms can be useful.
Authors: We thank the reviewer for these pertinent comments. The following sentence “(i) the stress responses common between amoebae and other organisms” has been changed into “The stress responses common in eukaryotes”. The order of sentences has been also modified according to the reviewer suggestions.
I am also confused about information included in the figures. From where is this information coming? If it is the result of your own study then the legends should include more details about methodology. If these are not your results then the clear reference should be included in the legends.
Authors: the images and the GO Terms search figure are from our own laboratories. Figure 1 was unpublished, figure 2 derive from published data but unpublished image and bioinformatics searches are novel. We have provided precisions in their legends.
More specifically:
Figure 3. Genome accession numbers must be indicated.
Authors: We retrieved the GO annotations from the three reference genome of the species mentioned in the legend: The taxon IDs were used for retrieving the GO term annotations in QuickGO. According to NCBI. For more clarity, we added the taxon IDs to the legend. Furthermore, a excel file containing all genes ID was added as supplemental Table 1
L70-80 in the Introduction: I am not sure if I understand this sentence, it is very abstract and not informative => Omics data show the regulation of protein abundance [14,15] and gene regulation (i.e., tRNA, anti-sense RNA, mRNA) are common to encystation and oxidative stress [1,16–19]. => I suggest to be here more specific, which proteins, which genes? Are you talking about gene up or down regulation?
Authors: we precise that proteins are more abundant, and transcripts are upregulated. The sentence was improved as follow " Omics data show the regulation of protein abundance (i.e., Hsp20 in A. castellanii or Hsp70, chitinase and cyst wall proteins in E. histolytica) [14,15] and upregulation of gene expression for RNAs (i.e., tRNA, anti-sense RNA, mRNA) are common to encystation and oxidative stress [1,16–19]"
L86-87: We give significant emphasis to protein post-translational modifications (PTMs), in particular those involving ubiquitin-like modifiers (UBLs), which importance in stress has been recently discovered => provide reference
Authors: the following reference has been inserted within the text (Dragana Ilic et al., 2022 Seminars in Cell & Developmental Biology).
L91. 2. Shared stress responses enabling cell survival => Shared? Why shared? May be better to say combined?
Authors: the sub-title has been modified as follow: stress responses common in eukaryotes.
L250 => is this about amoebae or in general?
Authors: the sub-title has been changed as follow: Ubiquitin, Ubiquitin-like protein modifications and proteasome activities in pathogenic amoebae.
Round 2
Reviewer 2 Report
The authors did not add the figures source in the legend of figures. This referee suggests including *unpublished or published in reference xxx)
Author Response
Reviewer 2
The authors did not add the figures source in the legend of figures. This referee suggests including *unpublished or published in reference xxx)
Authors: the legends have been modified accordingly